# The Advantage of Using an Optical See-Through Head-Mounted Display in Ultrasonography-Guided Needle Biopsy Procedures: A Prospective Randomized Study

**DOI:** 10.3390/jcm12020512

**Published:** 2023-01-08

**Authors:** Tadafumi Shimizu, Takaaki Oba, Ken-ichi Ito

**Affiliations:** Division of Breast and Endocrine Surgery, Department of Surgery, Shinshu University School of Medicine, Matsumoto 390-8621, Japan

**Keywords:** optical see-through head-mounted display, augmented reality, ultrasonography-guided intervention

## Abstract

An optical see-through head-mounted display (OST-HMD) can potentially improve the safety and accuracy of ultrasonography (US)-guided fine-needle aspiration. We aimed to evaluate the usefulness of an OST-HMD in US-guided needle-puncture procedures. We conducted a prospective randomized controlled study in which we compared the accuracy and safety of the US-guided needle puncture procedure and the stress on the practitioner when using OST-HMD versus standard US display (SUD). Inexperienced medical students were enrolled and randomly divided into two groups. A breast phantom was used to evaluate the required time and accuracy of the US-guided needle puncture. Practitioner stress was quantified using a visual analog scale (VAS). When the procedure was performed for the first time, the time required to reach the target lesion at a shallow depth was significantly shorter in the OST-HMD group (39.8 ± 39.9 s) than in the SUD group (71.0 ± 81.0 s) (*p* = 0.01). Using the OST-HMD significantly reduced the unintentional puncture of a non-target lesion (*p* = 0.01). Furthermore, the stress felt by the practitioners when capturing the image of the target lesion (*p* < 0.001), inserting and advancing the needle more deeply (*p* < 0.001), and puncturing the target lesion (*p* < 0.001) was significantly reduced in the OST-HMD group compared with that in the SUD group. Use of OST-HMD may improve the accuracy and safety of US-guided needle puncture procedures and may reduce practitioner stress during the procedure.

## 1. Introduction

Ultrasonography (US) plays an indispensable role in the diagnostic evaluation of superficial organs, such as the thyroid and breast, and fine-needle aspiration cytology (FNAC) or core needle biopsy is essential for the pathological diagnosis of the lesion [1,2,3]. US-guided insertion of a puncture needle is performed to approach the target lesion accurately; however, considerable experience is required to become proficient in this technique. One of the factors that makes this procedure difficult for beginners is that they cannot see the needle puncture site while looking at the monitor of the US machine. A US machine is generally placed next to the patient. During the needle biopsy procedure, practitioners have to check the puncture site adjacent to the US probe and monitor by alternately turning and twisting their heads or bodies while holding the US probe (Figure 1a,b). The probe easily slips owing to the combination of the ultrasound gel beneath it and needle insertion. The repetition of such head or body movements may cause unintentional hand movements of practitioners, which may compromise the accuracy and safety of the procedure. Furthermore, the procedure’s complexity may increase the psychological stress experienced by practitioners. In addition, because serious complications, such as hematoma due to mispuncture of blood vessels in FNAC for thyroid nodules and pneumothorax in FNAC for breast tumors can occur, the development of a safer ultrasound-guided needle puncture technique is essential for medical safety [4,5]. Augmented reality (AR) is a technology that presents virtual information to a user while viewing a direct view of the real world [6]. Among many AR devices, optical see-through head-mounted displays (OST-HMDs) have recently attracted considerable attention in the medical field [7]. An OST-HMD enables practitioners to see the images on a monitor just in front of their eyes with their hands free. Its advantages have been reported in central venous catheterization, spinal puncture, and orthopedic and neurosurgery [8,9,10,11,12,13,14,15,16,17,18]. Thus, when the OST-HMD is applied to US-guided needle biopsy procedures, the practitioner can perform the insertion of the needle without moving their heads or bodies, which is expected to make the puncture procedure more accurate and safer. However, only a few studies have evaluated the utility of OST-HMD in US-guided puncture procedures for the diagnosis of thyroid and breast lesions.

To verify the benefit of OST-HMD in the US-guided needle puncture procedure, we conducted a prospective randomized controlled study in medical students with no experience in US-guided procedures in a preclinical model using a breast phantom.

## 2. Materials and Methods

### 2.1. Study Design

We conducted a prospective randomized study to assess the utility of an OST-HMD in the US-guided needle puncture procedure in a preclinical model using a breast phantom. Medical students assigned to the Division of Breast and Endocrine Surgery of Shinshu University School of Medicine in 2020 and 2021 for clinical clerkship were enrolled as practitioners and randomized into two groups at a 1:1 allocation ratio: a standard ultrasound display (SUD) group and an OST-HMD group. None of the participants had any experience with US-guided intervention. Randomization was performed by simple randomization using a random number. A total of 112 participants were included in the study. This study was approved by the Institutional Ethics Committee on Clinical Investigation of Shinshu University (no. 4885). Informed consent was obtained from all the participants.

### 2.2. Devices

We used a commercially available binocular OST-HMD, Moverio BT-35E (SEIKO EPSON Corporation, Suwa, Japan), with a resolution of 1280 × 720 pixels and a weight of 119 g (Figure 2a,b). Ultrasound examination was performed using a Noblus (Hitachi Medical Co., Ltd., Tokyo, Japan), with a resolution of 1024 × 768 pixels. Images were obtained with a 5–18 MHz linear array transducer (EUP-L75), and images were projected both on the display of the Noblus or on the Moverio simultaneously without delay via a mirroring adaptor connecting cable. A practitioner perceived the display of Moverio on a monitor of the same size as a 40-inch television screen floating 2.5 m in front of the practitioner. They could view the procedural site and their hands below the image projected onto the virtual screen. Consequently, they could see the procedural site and US image alternately by moving their eyes slightly up and down without turning their head.

A US-guided breast biopsy phantom (Kyotokagaku Co., Ltd., Kyoto, Japan) containing mock target lesions placed at three different depths was used to evaluate the required time and accuracy of the US-guided needle puncture (Figure 2c,d). Mock target lesions were placed at shallow (1 cm), intermediate (2 cm), and deep (3 cm) positions from the surface of the breast phantom. The target lesions in the intermediate position were 6 mm in diameter, whereas the others were 10 mm in diameter.

### 2.3. Ultrasonography-Guided Needle Puncture Procedure

The breast phantom was placed on an examination bed of height 60 cm, which is typically used in clinical practice. The US machine was placed on the right side of the bed. The horizontal distance between the display of the US machine and the breast phantom was fixed at 90 cm. The practitioner sat on the right side of the bed in front of the US machine. Each practitioner could adjust the chair’s height according to their preferences (Figure 3a).

After 10 min of pre-study practice supervised by two experts in US-guided intervention (T.S. and T.O.), the practitioners held a 22-gauge, 38 mm length needle with their right hands and a US probe with their left hands, regardless of their dominant arms, and started a needle puncture procedure. First, the practitioners were allowed to move the US probe freely over the breast phantom until they identified the image of the mock target lesion. When the practitioners were ready, they began inserting the needle under real-time US guidance using the assigned device (SUD or OST-HMD) (Figure 1a,b and 3b). With OST-HMD, the practitioners viewed the image on the virtual screen, as shown in Figure 3c. The practitioner first punctured the shallow target lesions. Immediately after the practitioners punctured the needle into the shallow target lesion, they removed the needle and placed it on the examination table. The practitioners performed the same procedure on the phantom’s intermediate and deep target lesions. An independent observer (T.S. or T.O.) measured the procedural time from when a practitioner held the needle to when they punctured each target lesion, even if the needle hit the targeted lesion only slightly. When the observer could confirm that the puncture needle hit the target lesion in the breast phantom by watching the display of Noblus, the puncture was considered successful. If a practitioner could not puncture the target lesion within 5 min, the result of the procedure was judged as a failure. If a practitioner lost sight of the tip of the needle on the SUD or OST-HMD monitor and unintentionally punctured a non-target lesion, the procedure was judged as an unsafe puncture. The frequency of unsafe punctures was used as an indicator of safety. Unsafe punctures were included as failures.

### 2.4. Data Collection

The characteristics of the practitioners, including age, sex, and the side of the dominant arm, were collected. As an index of ease of the procedure, the time required to puncture each target at three different depths (shallow, intermediate, and deep) was recorded for each practitioner, and the total procedure time was calculated. The number of unsafe punctures for each practitioner was used as the safety index. After performing the needle puncture procedure using the assigned device (SUD or OST-HMD), each practitioner performed the same procedure using another device. After the practitioners experienced both procedures, they were asked to voluntarily fill out the stress felt at the following four time points during the procedure using the visual analog scale (VAS) (Appendix A): (1) when identifying the image of the target lesion on the monitor, (2) insertion of the needle into the breast phantom, (3) when the needle was advanced into the phantom, and (4) when the target lesion was punctured.

### 2.5. Statistical Analysis

Categorical variables were analyzed using the chi-square test, while continuous variables were analyzed using the Mann–Whitney *U* test or two-sided paired *t*-tests. All statistical analyses were carried out using GraphPad Prism 8.0.2 (GraphPad Software, San Diego, CA, USA), and *p* < 0.05 was considered to indicate statistical significance.

## 3. Results

### 3.1. Participant Characteristics

One hundred and twelve medical students were enrolled in this randomized study (Table 1). The mean age of the participants (± standard deviation) was 23.9 ± 2.9 years. Seventy-eight (69.6%) patients were males, and 34 (30.4%) were females. One hundred and six participants (94.6%) were right-handed, and six (5.4%) were left-handed. The participants were randomized into two groups: SUD (*n* = 57) and OST-HMD (*n* = 55). There were no significant differences in age (*p* = 0.96), sex (*p* = 1.00), or the side of the dominant hand (*p* = 0.68) between the two groups.

### 3.2. Comparison of Procedural Time, Number of Failed Procedures, and Frequency of Unsafe Punctures between the SUD and OST-HMD Groups

A comparison of the time required for the puncture of the target lesions in the breast phantom and the number of unsafe punctures between the SUD and OST-HMD groups is presented in Table 2 and Figure 4. The time required to puncture the shallow target lesion was significantly shorter in the OST-HMD group (39.8 ± 39.9 s) than in the SUD group (71.0 ± 81.0 s) (*p* = 0.011) (Figure 4a). The time required to puncture the intermediate or deep target lesions tended to be shorter in the OST-HMD group than in the SUD group. However, the difference was not significant (intermediate: SUD 116.7 ± 107.8 vs. OST-HMD 105.0 ± 106.6 s, *p* = 0.61, deep: SUD 134.5 ± 119.0 vs. OST-HMD 125.3 ± 108.7 s, *p* = 0.97) (Figure 4b,c). The total procedure time in the OST-HMD group (322.2 ± 225.4 s) was shorter than in the SUD group (270.1 ± 193.5 s), although the difference was not statistically significant (*p* = 0.28) (Figure 4d). The number of failed procedures was higher in the SUD group (15.2%) than in the OST-HMD group (12.1%), although the difference was insignificant (*p* = 0.43). Furthermore, the frequency of unsafe punctures was significantly lower in the OST-HMD group (1.2%) than in the SUD group (7.5%) (*p* = 0.011).

### 3.3. Assessment of Practitioner Stress during the Ultrasound-Guided Needle Puncture Procedure

Comparisons of the VAS scores evaluating the stress felt by the practitioners while performing the US-guided needle puncture procedure between the SUD and OST-HMD groups are shown in Table 3 and Figure 5. VAS scores were assessed at four time points during the needle puncture procedure. VAS scores were significantly lower in the OST-HMD group than in the SUD group at each time point. Thus, the stress felt by the practitioner during the US-guided puncture procedure was significantly reduced by using the OST-HMD compared with the stress experienced in performing the procedure by viewing the standard US monitor.

## 4. Discussion

In clinical practice, invasive procedures are unavoidable and are performed in various situations to diagnose disease. Ideally, an accurate diagnosis should be made while adopting all possible safety measures to avoid complications from examination procedures. To safely perform invasive examination procedures, gaining experience and proficiency in the examination is crucial. However, developing more ergonomic and user-friendly examination equipment is also essential to enhance safety. US-guided needle biopsy techniques are essential for diagnosing breast, thyroid, and liver lesions. However, owing to various complications from the needle puncture, which can sometimes be serious, proficiency is required to perform the procedure safely.

Practitioners generally perform US-guided needle insertion by a freehand technique, in which they must insert a needle in the target with one hand while holding the probe with the other for adequate visualization; guidance under freehand conditions is often challenging and time-consuming for novice practitioners. It has been reported that using a needle guidance system in US-guided needle puncture procedures reduces the time required for the procedure for both expert and inexperienced radiologists [19,20]. However, Kaji et al. reported that novices (medical students) looked away from the US monitor more frequently than residents looked away in real-time US-guided central venipuncture [21], indicating the difficulty experienced by a novice when performing a needle puncture without visualizing the puncture site while holding the handheld probe. Therefore, a method to perform US-guided puncture procedures while looking at the hand would aid in performing the procedure more safely and in a shorter time.

OST-HMD, an AR technology, is anticipated to be useful when applied to various medical procedures because it can present images necessary for performance of a task in front of the operator’s eyes. Although it is also anticipated to be useful when applied to ultrasound-guided needle biopsy procedures, no studies have evaluated its usefulness in randomized controlled trials in beginners.

One study validated the usefulness of OST-HMD in US-guided needle biopsy procedures [22]. In this study, two skilled otolaryngologists performed US-guided FNAC using an OST-HMD or a standard US monitor during a US-guided needle puncture procedure in patients assigned to the two groups. They then compared the practitioner’s fatigue and procedural time between the two groups. The results showed that the use of the OST-HMD reduced fatigue among the practitioners. However, the OST-HMD and the standard monitor did not differ in procedural time. As the practitioners in that study were two otolaryngologists with sufficient clinical experience in US-guided FNAC procedures with a standard US monitor, the results observed are likely to differ from those observed for beginners in ultrasound-guided needle procedures. The present prospective randomized controlled study was performed on medical students with no experience with US-guided examination procedures, demonstrating the benefit of using OST-HMD in US-guided puncture procedures in medical students. To the best of our knowledge, this is the first study to demonstrate the advantage of using OST-HMD compared to SUD in terms of procedure time and safety in the needle puncture procedure in a prospective randomized study with many beginners, albeit in a preclinical setting.

As we enrolled medical students with no prior experience in ultrasonography as practitioners in this study, we could evaluate the utility of the OST-HMD in performing US-guided needle puncture procedures without bias. The practitioners were able to puncture the target lesion placed at a shallow position more quickly and safely using the OST-HMD than the SUD. The results indicate that the OST-HMD may allow the practitioner to manipulate the probe and puncture needle without diverting their gaze from the hand, which may be advantageous for novices performing ultrasonography. However, for the puncture of targets in the intermediate and deep positions, there was no significant difference between the procedure time with OST-HMD and SUD, although the procedure time for the OST-HMD group was shorter than that for the SUD group. This result suggests that puncturing deeper targets is difficult for novices and requires training, even if they can see the probe and needle just beneath the US display. On the other hand, it has been reported that the mean distance from skin to the breast cancer measured by an ultrasound machine was less than 1 cm in breast cancer patients [23,24]. These findings imply that the advantage observed in our study in puncturing a “shallow” targeted lesion may be useful in clinical practice.

Another expected advantage of using OST-HMD is that it may reduce the practitioner’s stress during the procedure. Although most previous studies investigating the utility of OST-HMD focused on how it could improve the technical outcome and did not assess stress [9,11,13,25], we tested whether the stress experienced during the procedure, owing to users being novices in the use of ultrasonography, would change with the use of the OST-HMD. Our results showed that the stress felt by the practitioner was significantly reduced with the use of the OST-HMD compared with that while using the SUD in all steps of the ultrasound-guided needle biopsy procedure. These results indicate that novice practitioners are anxious about taking their gaze away from their hands during needle puncture and when capturing the target image with the US probe. Thus, the present study suggests that using the OST-HMD may make US-guided biopsy and ultrasonography less stressful for beginners.

There are two types of commercially available OST-HMD devices, monocular and binocular. In the medical field, monocular devices, such as Google Glass (Google, Mountain View, USA), ORA-2 (Optinvent, Rennes, France), and Vuzix M300 (Vuzix, Rochester, USA) have been reported to be useful for surgery [26]. In procedures using monocular OST-HMD, the screen is in front of only one eye. Hence, vision is not restricted compared to a binocular device, which is an advantage of the monocular device. However, the screen is generally located on the lateral side of either the right or left eye; thus, the operator must move their eyes to the display to see the projected image. These eye movements can cause stress. Furthermore, monocular vision is inferior to binocular vision for accurate images [27]. Another disadvantage of monocular devices is that the screen is smaller than that for binocular devices. Because clearer images are considered necessary for safe and accurate US-guided needle biopsy, binocular devices, such as the Moverio BT-35E used in this study, are more suitable for this medical procedure.

This study has several limitations. First, the target of the US-guided intervention in this study was a breast phantom but not a patient. Second, the practitioners enrolled in this study did not have experience with ultrasonography or other invasive medical examinations. Therefore, to demonstrate the usefulness of OST-HMD in a clinical setting, it is necessary to conduct a prospective study in which physicians who have completed their initial training would perform the procedure in patients with breast or thyroid gland tumors. Third, we only measured subjective stress in the practitioners, and no objective measurements were performed. In a subsequent study, we plan to perform objective stress measurements, such as of changes in practitioners’ heart rates during the examination procedure.

## 5. Conclusions

This prospective randomized study showed the potential of OST-HMD in reducing the procedure time for a specific condition, improving the safety of US-guided needle puncture procedures in novice practitioners, and reducing their stress during ultrasonography. Therefore, the OST-HMD may be a helpful device for introducing US-guided interventions to novice practitioners.

## Figures and Tables

**Figure 1 jcm-12-00512-f001:**
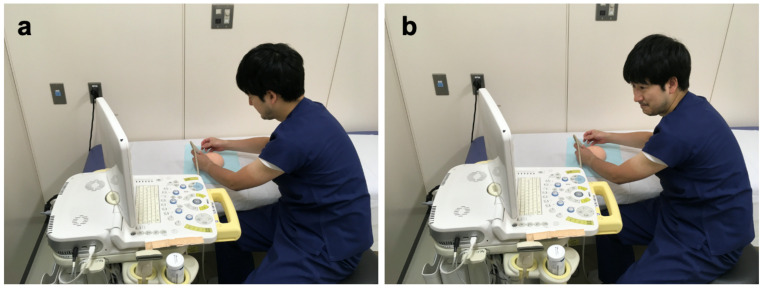
(**a**,**b**) Photographs during US-guided procedure with the standard US display (SUD).

**Figure 2 jcm-12-00512-f002:**
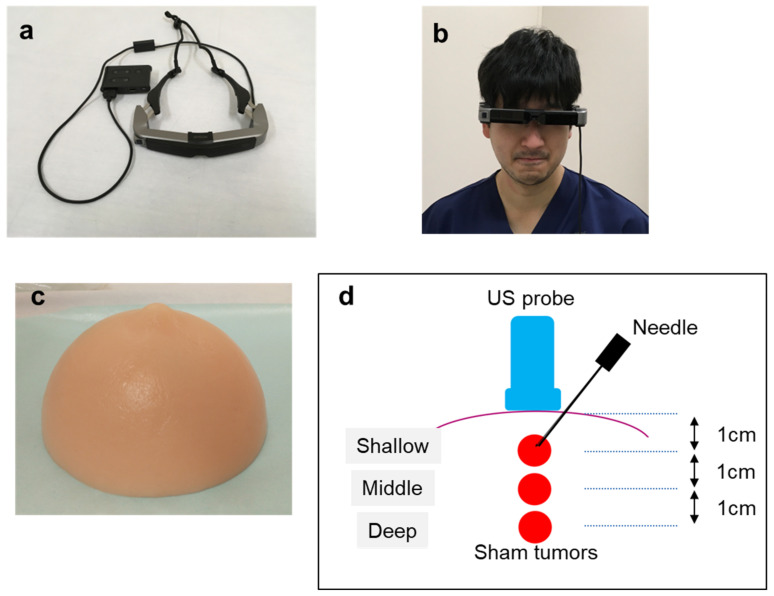
(**a**) OST-HMD (Moverio BT-35E, Seiko Epson Corporation, Suwa, Japan) used in this study. (**b**) Front view of a practitioner wearing OST-HMD. (**c**) Appearance of breast phantom (Kyoto Kagaku Co., Ltd., Kyoto, Japan) used in this study. (**d**) Schematic image of breast phantom. Three sham tumors were located at different depths with 1 cm intervals. Each tumor was referred to as a “shallow,” “middle,” or “deep” tumor.

**Figure 3 jcm-12-00512-f003:**
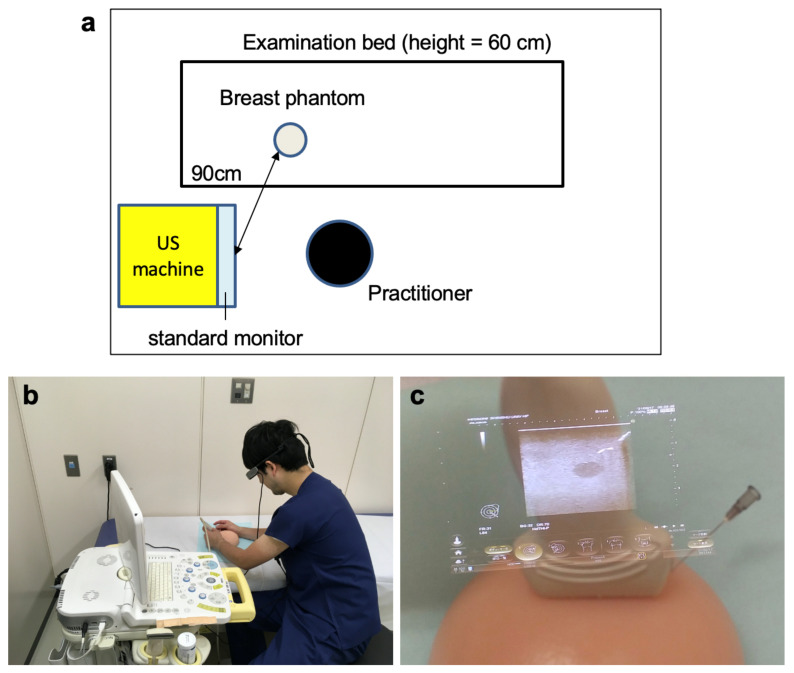
(**a**) Placement of the practitioner, US machine, and breast phantom during the procedure. The breast phantom was placed on the examination bed which was 60 cm in height. The US monitor was located at 90 cm distance from the breast phantom as indicated. The practitioner sat in front of the US monitor. (**b**) Photographs during US-guided procedure with the OST-HMD (**c**) A representative image projected on the OST-HMD when the procedure was performed. The view seen by the practitioner with OST-HMD. The practitioner can see the US image on the display of OST-HMD over the actual view of the procedural site.

**Figure 4 jcm-12-00512-f004:**
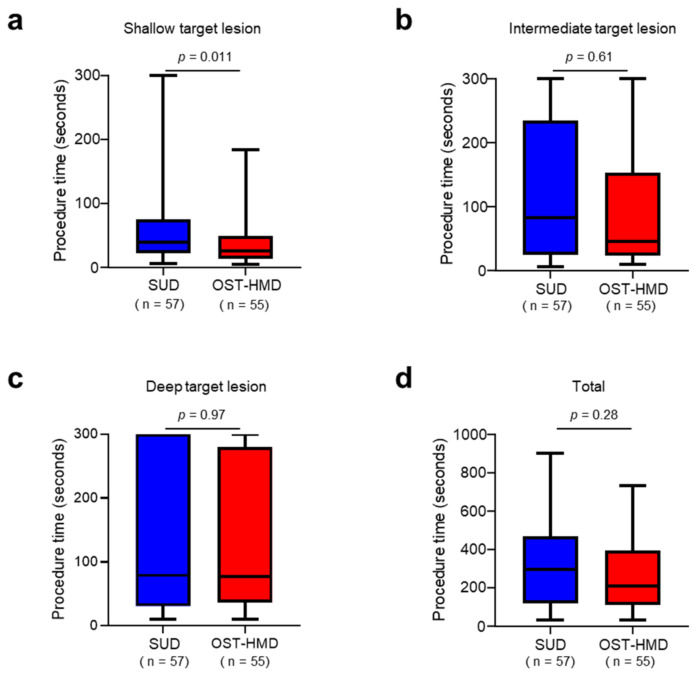
Box-and-whisker plot for the procedure time in the SUD group and the OST-HMD group for the shallow target lesion (*p* = 0.011) (**a**), the intermediate target lesion (*p* = 0.61) (**b**), and the deep target lesion (*p* = 0.97) (**c**). Total procedure time is shown in (**d**) (*p* = 0.28). *p* value was calculated using the Mann–Whitney *U* test. SUD: standard US display, OST-HMD: optical see-through head-mounted display.

**Figure 5 jcm-12-00512-f005:**
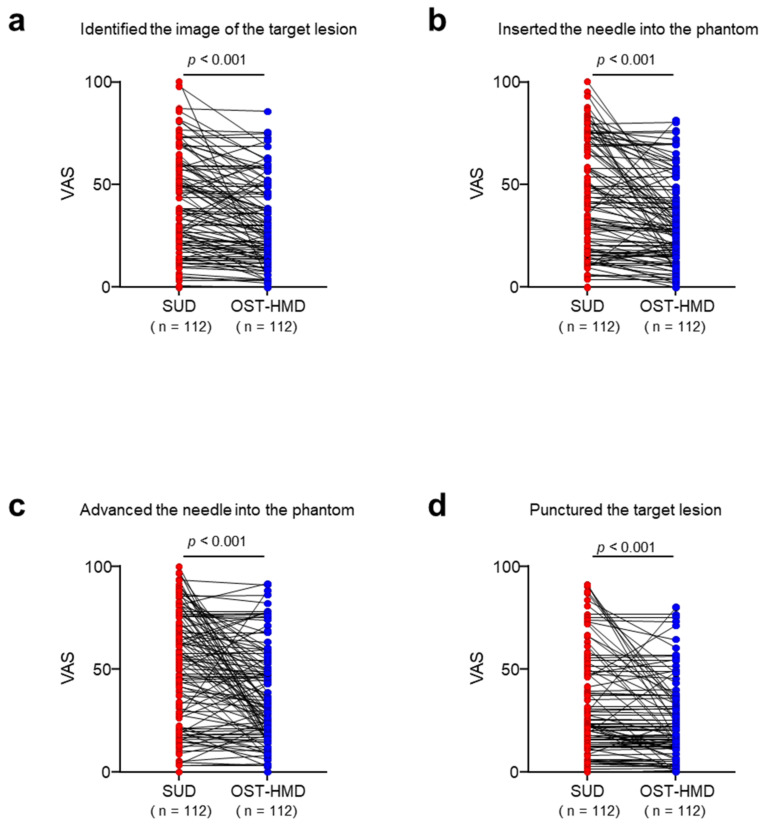
Stress quantified by VAS in the SUD method and OST-HMD method. *n* = 112 per group. *p* value was calculated with two-tailed paired *t*-test. VAS: visual analog scale, SUD: standard US display, OST-HMD: optical see-through head-mounted display.

**Table 1 jcm-12-00512-t001:** Characteristics of participants and comparison between the SUD and OST-HMD groups.

Variables	Total	SUD	OST-HMD	*p*-Value
	*n* = 112	*n* = 57	*n* = 55	
Age (mean ± SD)	23.9 ± 2.9	24.3 ± 3.5	23.5 ± 2.0	0.96
Sex				
Male	78 (69.6%)	40 (70.2%)	38 (69.1%)	1.00
Female	34 (30.4%)	17 (29.8%)	17 (30.9%)	
Dominant hand				
Right	106 (94.6%)	53 (93.0%)	53 (96.4%)	0.68
Left	6 (5.4%)	4 (7.0%)	2 (3.6%)	

SD: standard deviation, SUD: standard ultrasound display, OST-HMD: optical see-through head-mounted display.

**Table 2 jcm-12-00512-t002:** Comparison of the procedure time, the number of failed procedures, and the frequency of unsafe punctures between the SUD and OST-HMD groups.

Group	SUD	OST-HMD	*p*-Value
	*n* = 57	*n* = 55	
Procedure time (seconds, mean ± SD)			
Location of target lesion			
(1) Shallow	71.0 ± 81.0	39.8 ± 39.9	0.011
(2) Intermediate	116.7 ± 107.8	105.0 ± 106.6	0.61
(3) Deep	134.5 ± 119.0	125.3 ± 108.7	0.97
Total required time: (1) + (2) + (3)	322.2 ± 225.4	270.1 ± 193.5	0.28
Total numbers of failed procedures * (% of failed procedures **)	26 (15.2%)	20 (12.1%)	0.43
Total numbers of unsafe punctures * (% of unsafe punctures ***)	12 (7.5%)	2 (1.2%)	0.011

* Total attempts: SUD; 57 × 3 = 171, OST-HMD; 55 × 3 = 165, ** Total number of failed procedures × 100/total attempt number, *** Total number of unsafe punctures × 100/total attempt number, SD: standard deviation, SUD: standard ultrasound display, OST-HMD: optical see-through head-mounted display

**Table 3 jcm-12-00512-t003:** VAS scores evaluated by the practitioners while performing the US-guided needle puncture procedure.

Time Points	SUD	OST-HMD	*p*-Value
(1) Identified the image of the target lesion on the monitor	41.8 ± 25.4	32.7 ± 21.4	<0.001
(2) Inserted the needle into the breast phantom	43.8 ± 28.1	31.5 ± 22.0	<0.001
(3) Advanced the needle into the phantom	52.8 ± 27.8	39.7 ± 24.3	<0.001
(4) Punctured the target lesion	30.5 ± 27.4	23.6 ± 20.8	<0.001

SD: standard deviation, VAS; visual analog scale, SUD: standard ultrasound display, OST-HMD: optical see-through head-mounted display.

## Data Availability

The data supporting the findings of this study are available from the authors upon reasonable request.

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
