# Peer review of "The Advantage of Using an Optical See-Through Head-Mounted Display in Ultrasonography-Guided Needle Biopsy Procedures: A Prospective Randomized Study"

_jcm, 2023, doi:10.3390/jcm12020512_

Round 1

Reviewer 1 Report

This paper investigated the benefit of OST-HMD in the US-guided needle puncture procedure. The authors conducted a prospective randomized controlled study in medical students with no experience in US-guided procedures in a preclinical model using a breast phantom. The manuscript is well-written and easy to follow but lacks a sufficient review of existing technologies and key evidence to support the conclusions in the reviewer's judgment.

Introduction

1. The problem stated in this study is hand-eye coordination between the US machine and the needle. Other than OST-HMD, there are various kinds of devices, such as mechanical guidance and needle tracking/navigation systems, proposed to solve the problem and enhance the success rate. Before claiming the advantages of the OST-HMD, those technologies need to be reviewed and compared. 

2. Continue from 1, what was the improvement in the success rate that those existing technologies achieved? This should be compared/discussed in order to determine how beneficial the OST-HMD is.

2. A detailed review of the OST-HMD is needed to justify why it is worth a study to prove its effectiveness in assisting needle biopsy. Also, why is this specific brand of device selected? Does it represent the majority of OST-HMDs?

Methods

1. What's the reason for including only beginners? The stated problem seems to apply to all levels of practitioners. This limitation might have weakened the significance of the study.

2. In the caption of Figure 2, it might be better to replace "superficial" with "shallow" for consistency.

3. How to judge a successful puncture? The explanation is not clear enough. Was it judged by looking at the US machine or biopsy result?

Results & Discussion

1. Based on Table 2, the benefit of OST-HMD is unclear, except for shallow lesions and the total number of unsafe punctures, leading to weak support for the conclusion. This issue needs to be well-discussed.

2. Is the VAS score determined by the practitioner themselves, and in what method? Can details be provided?

Conclusions

Although the VAS scores showed a significant difference between the proposed and traditional methods, they were subjective evaluations. In addition, the operation time and success rate didn't show significant results, making it doubtful that the conclusion statement: "the OST-HMD may be an appropriate and useful device for US-guided interventions in clinical practice."

Author Response

Reviewer 1

This paper investigated the benefit of OST-HMD in the US-guided needle puncture procedure. The authors conducted a prospective randomized controlled study in medical students with no experience in US-guided procedures in a preclinical model using a breast phantom. The manuscript is well-written and easy to follow but lacks a sufficient review of existing technologies and key evidence to support the conclusions in the reviewer's judgment.

Introduction

  1. The problem stated in this study is hand-eye coordination between the US machine and the needle. Other than OST-HMD, there are various kinds of devices, such as mechanical guidance and needle tracking/navigation systems, proposed to solve the problem and enhance the success rate. Before claiming the advantages of the OST-HMD, those technologies need to be reviewed and compared. 

Reply:

We appreciate your insightful comments. When performing ultrasound (US) examinations using a handheld probe, novices often find it difficult to scan with a handheld probe while looking at the US monitor. Kaji et al. reported that novices (medical students) looked away from the US monitor more frequently than residents looked away in real-time US-guided central venipuncture (Kaji T, et al., J Vasc Access., 23:360-364, 2022), indicating the difficulty experienced by a novice when performing a needle puncture without visualizing the puncture site while holding the handheld probe.

As the reviewer pointed out, using a needle guidance system in US-guided needle puncture procedures reduces the procedural duration required by both expert and non-novice inexperienced radiologists (Bluvol, et al., Med Phys. 2008 35:617-28, Bluvol et al, AJR 2009; 192:1720–1725).

The needle guidance system is considered useful for examiners with some experience in US examinations using handheld probes. However, as shown by Kaji et al., novice examiners may be uncomfortable performing a needle puncture without looking at their hands.

Therefore, a method to perform US-guided puncture procedures while looking at the hand would aid in performing the procedure more safely and in a shorter time. Hence, we conducted a prospective study to verify whether the OST-HMD is useful in US-guided needle puncture procedures for those performing the procedure for the first time. We have added the several sentences in the “Discussion” to clarify this matter (line 233–247, page 10).

  1. Continue from 1, what was the improvement in the success rate that those existing technologies achieved? This should be compared/discussed in order to determine how beneficial the OST-HMD is.

Reply:

Bluvol et al. (referred in response to Query 1) reported a significantly higher success rate of the needle puncture procedure with a needle guidance system than that of the freehand technique (95.9% vs. 91.3%, p<0.05).

As they did not include novices with no previous US experience, their results may not be comparable to those of the present study, in which all the examiners were novices with no prior US experience.

We have referred to the results of that study in the “Discussion” section in accordance with your advice (line 233–247, page 10).

  1. A detailed review of the OST-HMD is needed to justify why it is worth a study to prove its effectiveness in assisting needle biopsy. Also, why is this specific brand of device selected? Does it represent the majority of OST-HMDs?

Reply:

Since only a few studies have examined the application of OST-HMD to US-guided needle puncture procedures, we deemed it necessary to prospectively evaluate its usefulness. We had described an overview of previous studies in the third paragraph of the Discussion section of the original manuscript, which we identified via a literature search.  

A limited number of binocular-type OST-HMDs are available in Japan. We used an OST-HMD manufactured by EPSON because it is the most readily and affordably available OST-HMD, and we could request the company for technical advice as we had acknowledged them in the Acknowledgement. We could not compare this OST-HMD with other OST-HMDs because no other OST-HMD with a high-resolution display is readily available in Japan. We humbly request the reviewer’s understanding.

Methods

  1. What's the reason for including only beginners? The stated problem seems to apply to all levels of practitioners. This limitation might have weakened the significance of the study.

Reply:

As we have mentioned in the response to Query 1 regarding the “Introduction,” novice US practitioners cannot easily scan with a handheld probe without looking at their hand. Especially in sloped body areas such as the breast, the handheld probe is often misplaced from the targeted lesion when the examiners look at the US monitor. However, as practitioners gain proficiency in ultrasonography, many find it easier to scan with a handheld probe while keeping their eyes on the US monitor. To evaluate the usefulness of the OST-HMD without an experience-based bias, we considered that novice medical students without experience using the handheld probe would be optimal candidates.

As pointed out by the reviewer, we agree that a verification with skilled practitioners is necessary to improve the future clinical application of the OST-HMD. We are planning a prospective trial with practitioners with different experience levels in ultrasonography. We had described this issue as a limitation of our study in the original manuscript, and have modified the "Conclusion" in the revised manuscript accordingly (line 322–340, page 11–12).

  1. In the caption of Figure 2, it might be better to replace "superficial" with "shallow" for consistency.

Reply:

We thank the reviewer for their careful reading of our manuscript. We have replaced "superficial" with "shallow" in the legend of Figure 2 (line 102, page 3). 

  1. How to judge a successful puncture? The explanation is not clear enough. Was it judged by looking at the US machine or biopsy result?

Reply:

We apologize for the insufficient explanation. When an observer could confirm that the puncture needle hit the target lesion in the breast phantom by watching the display of US machine (Noblus), the puncture was considered successful.

We have added this to the subsection “2.3 Ultrasonography-guided needle puncture procedure” in the “Materials and Methods" section (line 126–128, page 4).

Results & Discussion

  1. Based on Table 2, the benefit of OST-HMD is unclear, except for shallow lesions and the total number of unsafe punctures, leading to weak support for the conclusion. This issue needs to be well-discussed.

Reply:

The procedural duration for puncturing "shallow target lesions" was significantly shorter with the OST-HMD, while the procedural duration for puncturing "intermediate" and "deep" target lesions tended to be shorter with the use of OST-HMD, although no significant difference was observed. In this study, the practitioners were medical students with no prior experience with puncture procedures. The practitioners’ unfamiliarity with the puncture procedure might have influenced the results.

However, the frequency of unsafe punctures decreased significantly with the use of the OST-HMD, suggesting that its use reduces the incidence of losing the site of needle tip and increases the "safety" of the procedure.

The previous studies reported that the mean distance from the skin to the breast cancer measured by a US machine was less than 1 cm in breast cancer patients (Eom et al., J Surg Oncol. 2015;111:824-828, Ansari et al., Ann Surg Oncol. 2011;18:3174-80). These findings imply that the advantage observed in our study in puncturing a "shallow" targeted lesion at a depth of 1 cm may be useful in clinical practice. We have added some descriptions to clearly convey our thoughts in the “Discussion” section of the revised manuscript (line 287–291, page 11).

  1. Is the VAS score determined by the practitioner themselves, and in what method? Can details be provided?

Reply:

We apologize for the insufficient explanation. We have attached an English translation of the VAS (the original is in Japanese) that we used in this study as Supplementary Figure 1. We instructed each practitioner to voluntarily fill out this VAS score after completing the puncture procedure. We have added this information to the “Materials and Methods” section of the revised manuscript (line 150–153, page 5).

Conclusions

Although the VAS scores showed a significant difference between the proposed and traditional methods, they were subjective evaluations. In addition, the operation time and success rate didn't show significant results, making it doubtful that the conclusion statement: "the OST-HMD may be an appropriate and useful device for US-guided interventions in clinical practice."

Reply:

As noted by the reviewer, the Conclusions in the original manuscript went beyond the scope of our results. We have revised the content to limit or conclusions to those derived from the results of this study (line 335–340, page 12).

Reviewer 2 Report

The paper is very well written and contributes an Augmented reality (AR) scheme for ultrasonography (US)-guided fine-needle aspiration, which enables accuracy and safety. Furthermore, the proposed scheme outperforms the state of the arts and can be an appropriate and useful device for US-guided interventions in clinical practice.

There are some problems, which must be solved before it is considered for publication. If the following problems are well-addressed, this reviewer believes that the essential contribution of this paper is important for Biopsy Procedures.

1. The target lesion depth limitation

In Page 6 Figure 4, the authors demonstrated the procedure time in the SUD group and the OST-HMD group for the shallow (1 cm), intermediate (2 cm), deep (3 cm), and total target lesions. It decreased the procedure time at shallow target lesions, however, there is little difference at deep target lesions.

If the depth is the limitation of the optical see-through head-mounted display (OST-HMD), the authors should further demonstrate the actual depth of the target lesion for breast cancer patients, other than simplifying mimicking the shallow, intermediate, and deep at a breast phantom.

If the actual depth of the target lesion for breast cancer patients is commonly over than deep (3 cm), The advantage of the OST-HMD group will no longer exist. The reviewer suggests the authors could add some reference works of literature to demonstrate this issue.

2. Reduce stress limitation

In-Page 8 line 256, the authors said, “Our results showed that the stress felt by the practitioner was significantly reduced with the use of the OST-HMD compared with that while using the SUD in all steps of the ultrasound-guided needle biopsy procedure.” However, there are not enough experiments to explain the stress-reduced results.

The authors only studied the Procedure time, the Total number of failed/unsafe procedures, and different Time Points. The reviewer could not see any stress testing. If the authors want to show the reduction of stress, more experiments should be involved, such as stress echocardiograms.

3. Device limitation

In-Page 2 line 77, the authors described the device. However, more details should be discussed. For example, the resolution is 1280 × 720 pixels, it is clear enough to observe the very small target lesion?

The images were projected either on the display of the Noblus or the Moverio via a mirroring adaptor connecting cable. Is the image transmission fast enough and has no lag?

Author Response

Reviewer 2

The paper is very well written and contributes an Augmented reality (AR) scheme for ultrasonography (US)-guided fine-needle aspiration, which enables accuracy and safety. Furthermore, the proposed scheme outperforms the state of the arts and can be an appropriate and useful device for US-guided interventions in clinical practice.

There are some problems, which must be solved before it is considered for publication. If the following problems are well-addressed, this reviewer believes that the essential contribution of this paper is important for Biopsy Procedures.

  1. The target lesion depth limitation

In Page 6 Figure 4, the authors demonstrated the procedure time in the SUD group and the OST-HMD group for the shallow (1 cm), intermediate (2 cm), deep (3 cm), and total target lesions. It decreased the procedure time at shallow target lesions, however, there is little difference at deep target lesions.

If the depth is the limitation of the optical see-through head-mounted display (OST-HMD), the authors should further demonstrate the actual depth of the target lesion for breast cancer patients, other than simplifying mimicking the shallow, intermediate, and deep at a breast phantom.

If the actual depth of the target lesion for breast cancer patients is commonly over than deep (3 cm), The advantage of the OST-HMD group will no longer exist. The reviewer suggests the authors could add some reference works of literature to demonstrate this issue.

Reply:

We appreciate the reviewer’s valuable comments. As mentioned in the Discussion section of the original manuscript, as the practitioners were medical students with no prior experience with puncture procedures, their unfamiliarity with the puncture procedure might have influenced the results of the needle puncture of "Intermediate" and "Deep" tumors, which may be a reason why OST-HMD did not impart an advantage when puncturing intermediate and deep targets.

However, as suggested by the reviewer, the mean distance from the skin to the breast cancer measured by a US machine was 0.43-0.467 cm (n = 891) (Eom et al., J Surg Onco. 2015;111:824-828) or 0.91 cm (0.1-2.8 cm, n = 233) (Ansari et al., Ann Surg Oncol. 2011;18:3174-80) in breast cancer patients. These findings imply that the advantage observed in our study in puncturing a "shallow" tumor at a depth of 1 cm may be useful in clinical practice. We have discussed these details in the “Discussion” section (line 287-291, page 11).

  1. Reduce stress limitation

In-Page 8 line 256, the authors said, “Our results showed that the stress felt by the practitioner was significantly reduced with the use of the OST-HMD compared with that while using the SUD in all steps of the ultrasound-guided needle biopsy procedure.” However, there are not enough experiments to explain the stress-reduced results.

The authors only studied the Procedure time, the Total number of failed/unsafe procedures, and different Time Points. The reviewer could not see any stress testing. If the authors want to show the reduction of stress, more experiments should be involved, such as stress echocardiograms.

Reply:

As the reviewer pointed out, a limitation of the present study was that we only measured subjective stress in the practitioners, and no objective measurements were performed. In a subsequent study, we plan to perform objective stress measurements, such as measuring changes in practitioners' heart rates during the examination procedure. We have added this as a limitation of the present study in the “Discussion” section (line 329–333, page 12).

  1. Device limitation

In-Page 2 line 77, the authors described the device. However, more details should be discussed. For example, the resolution is 1280 × 720 pixels, it is clear enough to observe the very small target lesion?

The images were projected either on the display of the Noblus or the Moverio via a mirroring adaptor connecting cable. Is the image transmission fast enough and has no lag?

Reply:

The resolution of the display of the Noblus used in this study is 1024 x 768 pixels. As the resolution of the Moverio (OST-HMD) display is 1280 × 720 pixels, small target lesions can be easily identified. As the reviewer thought, the US image was projected on both the Noblus and Moverio displays simultaneously without delay via a mirroring-adaptor connecting cable. We have corrected and added this description of the device to the subsection “2.2 Devices” in the “Materials and Methods” section (line 83–83, page 2–3).

Round 2

Reviewer 1 Report

All my comments are well-addressed in this revision.

A minor question:

Why does every figure have a very large and bold title "Figure xx" on the upper left?